# Microbiota, Gut Health and Chicken Productivity: What Is the Connection?

**DOI:** 10.3390/microorganisms7100374

**Published:** 2019-09-20

**Authors:** Juan M. Diaz Carrasco, Natalia A. Casanova, Mariano E. Fernández Miyakawa

**Affiliations:** 1Instituto de Patobiología Veterinaria, Centro Nacional de Investigaciones Agropecuarias, Instituto Nacional de Tecnología Agropecuaria, Calle Las Cabañas y Los Reseros s/n, Casilla de Correo 25, 1712 Castelar, Buenos Aires, Argentina; diazcarrascojuan@gmail.com (J.M.D.C.); casanova.andrea@inta.gob.ar (N.A.C.); 2Consejo Nacional de Investigaciones Científicas y Técnicas, Godoy Cruz 2290, 1425 Ciudad Autónoma de Buenos Aires, Argentina

**Keywords:** poultry, gut health, intestinal microbiota, productivity, performance

## Abstract

Gut microbiota and its relationship to animal health and productivity in commercial broiler chickens has been difficult to establish due to high variability between flocks, which derives from plenty of environmental, nutritional, and host factors that influence the load of commensal and pathogenic microbes surrounding birds during their growth cycle in the farms. Chicken gut microbiota plays a key role in the maintenance of intestinal health through its ability to modulate host physiological functions required to maintain intestinal homeostasis, mainly through competitive exclusion of detrimental microorganisms and pathogens, preventing colonization and therefore decreasing the expense of energy that birds normally invest in keeping the immune system active against these pathogens. Therefore, a “healthy” intestinal microbiota implies energy saving for the host which translates into an improvement in productive performance of the birds. This review compiles information about the main factors that shape the process of gut microbiota acquisition and maturation, their interactions with chicken immune homeostasis, and the outcome of these interactions on intestinal health and productivity.

## 1. Introduction

The demand for poultry products has grown exponentially in recent decades and it is estimated that production will reach 130 million tons of chicken meat in 2020, when it will become the most consumed animal meat in the world [1]. Among traditional livestock species, poultry are the most efficient feed converters, with a feed conversion ratio in the range of 1.6–2.0. The maintenance of a high feed efficiency plays an essential role in the capacity of the poultry sector to meet the growing demand for poultry products, and it also represents a major challenge, since the intensive production of birds is very prone to be affected by outbreaks of infectious diseases, particularly in geographical areas where climatic conditions are changing.

The structure and functionality of the intestinal microbiota is crucial for the health of poultry since the process of acquisition and maturation of the gut microbiota throughout the growth cycle of the birds has a strong influence on the development of the intestinal epithelium and the modulation of the physiological functions required to maintain intestinal homeostasis (i.e., immunity, nutrient digestion, intestinal barrier integrity), and in turn these functions are essential to optimize the efficiency of extraction and use of energy by the host [2]. In this review, most of the information available about the factors that shape the gut microbiota of poultry regarding their interaction with the physiological functions of the host has been collected, with focus on the impact of these interactions on intestinal health and productive performance of the birds.

## 2. Key Factors that Shape Chicken Gut Microbiota

Intestinal microbiota is considered a crucial organ that plays an integral role in maintaining the health of the host by modulating several physiological functions including nutrition, metabolism, and immunity. The digestive process is strongly linked to gut microbiota; nutrient absorption, feed digestibility, energy harvest and therefore productivity are influenced by microbiota composition and diversity [3,4]. The chicken gut microbiota includes hundreds of bacterial species dominated at the phylum level by Firmicutes, Bacteroidetes, Proteobacteria and Actinobacteria [5,6]. The microbial communities differ through the chickens gut intestinal tract with particular microbial profiles detected in crop, gizzard, ileum, cecum and colon of broiler chickens [7,8,9,10].

The advent of *omics* and multi-*omics* approaches as tools for the study of microbial communities has allowed a detailed characterization of the gut microbiota of chickens in a quick and robust fashion, without the need to cultivate the microorganisms that compose it, as well as describing the alterations suffered by the gut microbiota in function of different stress factors of infectious, environmental and genetic nature [11,12]. Although certain variables have been analyzed in greater depth due to their direct influence on chicken productivity, there are other factors associated with the productive environment that have been less studied and can significantly alter the structure and functionality of the microbiota and therefore chicken’s health and productive performance (Figure 1).

### 2.1. Acquisition and Maturation of the Intestinal Microbiota

The initial colonization of the gastrointestinal tract of birds occurs naturally from the moment of hatching and can even begin before, by passing of microorganisms through the pores of the eggshell [13,14]. The intensive production of birds implements very strict hygiene practices that strongly reduce the microbial load in the hatching environment in order to avoid colonization by pathogenic bacteria, and so that the newly hatched birds acquire their initial microbiota at from an artificial environment on the farm instead of the natural maternal source [15]. However, available evidence also supports the hypothesis that part of the microbial colonizers in early embryos can be inherited from maternal hens that could be also adjusted by environmental factors (including immune system interaction) during different developmental stages [16]. Once hatched, the gastrointestinal tract of chickens becomes successively colonized by Enterobacteriaceae at first days of age and then Firmicutes (approximately from 7 days of age) [17]. However, colonization of gastrointestinal tract with specific bacterial species, belonging to the Enterobacteriaceae or Firmicutes groups, is likely a stochastic process driven by the contact with microorganisms coming from the rearing environment and from bacteria present in food and water [18].

There is consensus in the scientific community that early colonization of the intestine is of great importance for poultry health and productivity, since it can alter the morphology and physiology of the intestine and susceptibility to infectious diseases [2]. The inoculation with prebiotics, probiotics and consortia of microorganisms applied directly on or inside the egg are being evaluated as strategies to favor the early intestinal colonization of birds with a “healthy” microbiota from the moment of hatching [19,20,21].

After the initial colonization of the intestine a succession of microorganisms is observed in which the species richness and complexity of the population structure of the microbiota increase as the birds grow, until eventually microbiota reaches a state of maturation and stabilizes. This process normally occurs in commercial broiler chickens around 3 weeks of life [5,12,22,23]. However, although this is the general rule in commercial broiler chickens, development times and succession patterns of intestinal microbiota species can vary greatly depending on the genetic background of the birds and farm management factors [9,16,24,25,26,27,28]. For example, in laying hens four different stages of development of the cecal microbiota during the first year of life have been described, with significant changes in the cecal microbiota composition [29]. It has been reported that the succession of changes in the gut microbiota correlates with changes in the cytokine profile expressed by host intestinal cells in response to different bacterial groups [30]. The increase in the phylum Proteobacteria, which includes many potentially pathogenic bacteria, correlates with a pro-inflammatory cytokine profile, while the increase in members of the phylum Firmicutes is associated with an anti-inflammatory state. Therefore, gut microbiota is involved in the immune homeostasis of the gastrointestinal tract of birds, and therefore an imbalance in the intestinal microbiota can lead to an immune imbalance and an impact on birds’ health.

### 2.2. Influence of Climate and Seasonal Changes

The intensive breeding of chickens requires a rigorous control of the conditions in the commercial establishment to optimize the growth of the birds, mainly the temperature and the relative humidity [31]. However, even when regulating the internal variables of the poultry house, external climatic conditions, particularly extreme heat, can negatively affect chicken health and productivity [32,33]. In a recent study in a broiler farm from Argentina, we observed that there is a strong variation of the cecal microbiota according to seasons, with the highest species richness in summer, doubling that of winter [34]. This agrees with findings of other authors that observed a similar pattern of seasonal variation in alpha diversity when analyzing cecal samples from multiple flocks of broiler chickens in the U.S. [35]. On the other hand, many studies reported differences in the gut microbiota profiles of poultry according to geographic location [26,36,37]. Probably these variations can be partially attributed to the impact of regional and seasonal climatic conditions on the microbiota surrounding the birds and the birds itself. Considering influence of these climatic and environmental factors is essential when designing and conducting trials about the relationship between the intestinal microbiota and productive performance. Ignoring this information could induce an important bias and lead to erroneous conclusions.

### 2.3. Influence of Management Factors and Internal Conditions of the Farm

There are many factors associated with the internal management and decision-making of intensive poultry production establishments that are relevant for the establishment of a healthy intestinal microbiota. The influence of some of these factors on productive performance of broiler chickens has been widely studied [38], but there are still few studies that analyze these relationships based on the composition and diversity of the intestinal microbiota. Among the most relevant farm management factors, stand out the production system, particularly the choice of alternative production systems with access to range [4,39,40]; the hygiene and biosecurity programs, the protocols and criteria used to define the time of feed access, feed processing, and feeding programs [41,42,43], the protocols and criteria used to define the vaccination schemes, therapeutic medication and rotation of the antibiotic growth promoters throughout the year, factors associated with the stress and welfare of the birds, such as the stocking density [44,45], duration of photoperiod [46,47], ventilation and ammonia concentration in the air [48,49] and heat-stress [50,51]. It is clear that management decisions define the microbiota in the environment that surrounds the birds and therefore the intestinal microbiota, but more studies are needed under commercial conditions to determine the key points that contribute to maintaining a healthy intestinal microbiota and maximize chicken productivity.

### 2.4. Interplay between Intestinal and Litter Microbiotas

The litter on which the chickens are raised is usually composed of pine shavings rice hulls and other vegetal materials, and by feed, water, and chicken excreta which is mixed and composted with the bedding materials. The materials used as substrate for bedding can alter the morphology and physiology of the intestine and the composition of the intestinal microbiota [52]. Throughout the growth cycle, birds continuously peck and ingest litter particles and thus acquire an important part of the microorganisms that make up the intestinal microbiota. In turn, the litter accumulates fecal matter and constitutes a microbiota with its own diversity and composition [53]. In a recent work we performed a 1-year follow-up during six consecutive growth cycles of broilers and we observed a close correlation between cecal and litter microbiota, with the same seasonal pattern of variation in alpha and beta diversity in both environments [34]. Although the population structure is very different, certain groups of bacteria are shared and showed the same pattern of variation in both environments. Other authors described previously that litter microbiota correlates with the corresponding intestinal microbiota both in chickens and turkeys [4,54]. Furthermore, our results showed that seasonal changes in litter microbiota correlates with seasonal changes in flock productivity, which is better in winter and worsens in summer in conjunction with an increase in species richness, and the same patterns is observed in cecal samples [34]. This interrelation reflects the constant exchange of microorganisms that normally exists between the microbiota of the farm environment and the gut microbiota of broiler chickens, and highlights the role of litter as a reservoir of microbial diversity.

Reusing litter during consecutive flocks is a practice that is usually applied to reduce production costs of aviculture in many countries. It has been reported that this practice can alter the microbiota in the litter itself, where the prevalence of halotolerant/alkaliphilic bacteria is increased, but also altered the intestinal microbiota of the birds, increasing the levels of *Faecalibacterium prausnitzii*, a commensal butyrate-producing species, in the cecum of young chickens [55]. Other authors demonstrated that the load of intestinal bacteria is higher in reused litter and in beds with higher moisture content suggesting that it could imply a health risk with respect to the transmission of pathogens between flocks through the litter [56,57]. Although this risk is theoretically reasonable, more studies are needed to understand how the balance between commensal, probiotics, and pathogenic bacteria in the litter may affect chicken gut health.

## 3. Influence of the Microbiota on Immunity and Gut Health

The intestinal health of poultry has broad implications for the systemic health of birds, animal welfare, production efficiency, food safety, and environmental impact. Although the association between animal performance and gut health is widely accepted, there is not a clear definition of gut health itself. The concept of gut health includes morphological integrity, physiological functions of the intestinal tract (including digestion and absorption of nutrients), tissue metabolism, and energy balance, developed barrier functions, efficient immune responses, sustained inflammatory balance and, particularly, an adequate microbiota. On the other hand, health can be defined as the overall condition of an animal at a given time. Diseases produce a decline in health condition resulting in poor productivity and reduced quality of the affected animals, even lead to the death.

### 3.1. Interactions between Gut Microbiota and Chicken’s Immune System during Development

While much of the available data regarding the intestinal microbiota and enteric health relate to changes associated with a diseased state, there are clear examples of the importance of the microbiota in maintaining gut health and normal intestinal function. Several studies have focused on understanding the relationships between compositions of the native gut microbiota and colonization of the gut by pathogens. Gastrointestinal microbial community behaves as an anti-infectious barrier by inhibiting the pathogens’ adherence and subsequent colonization and the production of bacteriocins and of other toxic metabolites, excluding pathogenic microorganisms, fermenting complex polysaccharides, and providing energy to the host. Also, the initial interactions between gut microbial community and host innate immune system can lead to subsequent adaptive immune response, which can either be B-cell dependent or T-cell dependent [53]. Many host factors can influence the establishment and evolution of the gastrointestinal microbiota, but the gut microbiota–immune system interactions are clearly dominant at some stages. Infections of the gastrointestinal tract are detected by the host immune system which then responds via a complex interconnecting system of pathways involving the innate and adaptive immune systems.

Beta-defensins are small antimicrobial peptides that are present on the intestinal epithelial surface and constitute an important part of innate immunity [58]. These peptides are produced in response to challenge by lipopolysaccharide, involving Toll-like receptors (TLR-4) and the transcription factors NF-κB [59]. Defensin expression in the gut is affected by microbial exposure and reduction in microbial exposure resulted in lower expression of duodenal and ceca beta-defensins at early growing period of birds [60]. These initial responses of host innate immune system to determined components of the gut microbial community can shape microbiota composition and lead to subsequent adaptive immune response, which can either be B-cell dependent or T-cell dependent. Furthermore, microbial community helps in excluding pathogens, fermenting several substrates and providing energy to the host [53].

Gastrointestinal tract harbors diverse and complex microbial communities and, therefore, extensive and complex interactions occur between the microbial cells and host immune system. The fate of these interactions is partially determined at the very beginning development stage of chicken embryo. Early microbiota colonization is also essential for healthy gut development. Initial studies with germ-free chickens have shown that microbial exposure is necessary for an appropriate development and maturation of the gut immune system [61]. Germ-free animals are more susceptible to intestinal infections, showing reduced vascularity, digestive enzyme activity, muscle wall thickness, cytokine production, and serum immunoglobulin levels, smaller Peyer′s patches and fewer intraepithelial lymphocytes, but increased enterochromaffin cell area [62]. It is considered that development and maturation of the immune system is highly dependent of antigen encounter. Comparison of germ-free chickens with conventional birds revealed that at 7d of age, the absence of microbes in the intestine affected neutral and acidic goblet cell number and density, sialylated and sulfated acidic mucin staining and the major intestinal mucin expression MUC2 [63], suggesting a less mature intestinal mucosa in germ-free animals. Therefore, the influence of the gut microbiota on immune system development is considerable and it would have potential implications for optimal health and productivity, considering energy saving, immunity, and inflammation. It has to be emphasized that every modification of the microbiota influences the health and metabolic status of the host, which in turn, influence microbiota and, in this manner, the host and microbiota are unified in infinite symbiosis.

### 3.2. Gut Microbiota Composition and Immune Homeostasis

Modification of gut microbiota in early life of chickens using oral antibiotics has been associated with subsequent immune dysfunction, including development of autoimmune diseases [64]. Changes in the host intestinal inflammatory response, as well as the structure and diversity of the gut microbial community, also occur when antibiotics are introduced into animal diets [65,66,67,68,69,70,71,72,73,74,75,76]. Also, antibiotic treatments have been shown to increase the prevalence and the susceptibility of the host to intestinal pathogens [64,77]. These antibiotics at low concentrations have been commonly used in the poultry industry all around the world as growth enhancers. A metabolomics approach used to characterize and identify the biochemical compounds present in the intestine of broiler chickens fed a standard diet supplemented with antibiotic growth promoters, virginiamycin or bacitracin methylene disalicylate, demonstrated that antibiotic supplementation had profound effects on the levels of a wide variety of chemical metabolites, particularly amino acids, fatty acids, nucleosides, and nicotinamide-related compounds [78] supporting the idea of an anti-inflammatory effect of antibiotic growth promotion supplementation. The changes would affect the production of peptide mediators of inflammation (adhesion molecules, cytokines etc.) [79]. Bacitracin methylene disalicylate fed birds showed microbial diversity and species richness similar to control animals but the composition of cecal and ileal bacterial communities were differentially affected by bacitracin [77,80]. Bacterial phyla were also not affected by bacitracin in feed, but bacterial population changes were observed at lower taxonomic level [77,80]. However, these changes in bacterial communities are correlated with cytokines gene expression (i.e., IL-10, IL-4, IFN-γ), particularly at early stages of growth. Cytokine production can be modulated by commensal bacteria in the gastrointestinal tracts [81] with a significant role in both innate and adaptive immune responses.

The innate immune response leads to subsequent adaptive immune response and stimulates the production of Th2 cytokines [82]. Increased expression of IL-4 and IL-10 in the early growth phase which decreased at later stages was observed in chickens [77]. These changes can potentially be attributed to changes in microbial communities due to age, antibiotic administration and to pathogens infection as *Campylobacter* or *Salmonella* [77]. For example, cytokine expression in response to infection by *Campylobacter jejuni* in chickens challenged at late days show up-regulation of IL-6, IL-17A, and IL-17F [83,84]. The timing of exposure to *C. jejuni* alter the cytokine expression profile throughout the growth cycle and in turn this immune response can drive shifts in gut microbiota composition [83]. Similarly, a correlation between cytokine profiles of chickens and their resistance to the infection by *Salmonella* Typhimurium was described [83]. However, also changes in other members of the microbiota as *Escherichia coli* were shown to correlate with higher susceptibility to colonization by *Salmonella* [85], and *Lactobacillus* isolates that were able to control *Salmonella* infection in chicken [86]. These examples illustrate the complexity of the interactions that exist in the chicken gut microbiota and how difficult is to define specific effects of determined gastrointestinal microorganisms.

Not only pathogens but different commensal members of gut microbiota are able to alter the immune homeostasis. Withanage et al. reported that the expression of IL-6 increased in the cecal tonsils, ileum, and spleen on day 1 and 3 p.i. when one-day-old chickens were infected with *Salmonella* Typhimurium [87]. Many Gram-negative pathogens such as *E. coli*, *Shigella*, and *Salmonella* are Proteobacteria with recognized pro-inflammatory mechanisms. Although no specific genera were significantly correlated with cytokine expression, the most abundant genera within phylum Proteobacteria were classified as *Escherichia*/*Shigella* [30]. Among Firmicutes, the modulation of cytokine expression varies depending on the species being analyzed. Several genera, including genus *Faecalibacterium*, were inversely correlated with the expression of pro-inflammatory cytokine (IL-1β, IL-18, IL-6) and positively correlated with anti-inflammatory (TGF-β4) cytokine expression, while other Firmicutes were positively correlated with the expression of pro-inflammatory cytokines [30]. The relative abundance of phylum Bacteroidetes, which usually represents the second most abundant phylum in cecal microbiota after Firmicutes and is dominated by genus *Bacteroides*, differently correlated with anti-inflammatory cytokines according to the age of the birds [30].

## 4. Relationship between Microbiota Composition and Productive Performance

The productivity of poultry can be measured through different parameters including feed conversion ratio (FCR), residual feed intake (RFI), body weight gain (BWG), apparent metabolizable energy (AME) and time to achieve market weight [3]. FCR is the most frequently used in terms of growth performance and is calculated as the ratio of feed consumed to weight gained. The factors that impact on microbiota composition can also modify the productivity and may explain, at least in part, the variability in performance observed in each flock.

### Microbial Diversity and Performance

The relationship between the diversity of species in the intestinal microbiota and the feeding efficiency of poultry is not clear. In cattle, a less diverse rumen microbiota was found in more efficient animals, and the authors hypothesized that this relationship is associated with specialization of the microbiota in the production of output metabolites which are more relevant for the hosts’ energy metabolism, mainly short chain volatile fatty acids that can be used as source of energy and carbon for the growth of the animal [88]. In line with this concept, some authors have reported a lower species richness in the intestine of chickens with greater feeding efficiency, but not when feces samples were analyzed [36,89]. However, several studies found that bacterial diversity within the intestinal tract is higher in birds with lower FCR or high feed efficiency [3,10,90,91,92].

Each section of the chicken gastrointestinal tract has a different profile of microorganism, though they are closely related. The results of microbiota studies are influenced by intrinsic characteristics of birds such as genetic, gender, age, breed, health status, as well as by farm conditions including type of diet, feed additives, environment, and farm management, among others. Table 1 summarizes the microbial taxa associated with high and low productivity in different gut sections of the chicken’s intestine.

While in most trials there were shifts in alpha and beta diversity, these changes were driven by different microbial taxa. In the small intestine, previous studies have not detected differences in the microbiota of birds with high or low performance in jejunum samples [91,99]. In the ileum, the detection of different bacterial taxa has been associated with chicken productivity. Contrary to expectations, two studies have associated the presence of enterobacteria with higher productivity and the presence of lactobacilli with low productivity (Table 1). Though this genus has been widely associated with birds’ healthier status and high productivity, low diversity could lead to the overgrowth of certain bacteria linked to the development of diseases [39]. The cecum is the gut section with the highest bacterial diversity and where more differences were detected in relation to chicken feed efficiency. Potential performance-related phylotypes were assigned to some bacteria species such as *Lactobacillus salivarius*, *Lactobacillus aviarius*, *Lactobacillus crispatus*, *Clostridium lactatifermentans*, different members of family Ruminococcaceae, *Bacteroides vulgatus*, *Akkermansia*, and *Faecalibacterium*, among others [10,90,93]. The abundances of *Escherichia*/*Shigella* were found to correlate negatively with growth and fat digestibility in broiler chickens [94]. Also, *Campylobacter* colonization of broiler chickens has been connected with reduced economic performance in terms of an increase in cumulative FCR [95,100]. In another work, repeated trials in separate flocks were performed to compare cecal microbiota composition of high and low performing birds [96]. The authors detected differences between bacterial groups associated with FCR. Moreover, each productivity outcome correlated differently with specific phylotypes even under controlled conditions. However, in all cases, *Lactobacillus* correlated with high FCR whereas the genus *Faecalibacterium* correlated with low FCR. In more recent studies, restrictive feeding approach was followed to clarify the impact of feed intake on ileal and cecal microbiota and physiological and functional characteristics in broiler chickens with different residual feed intake as measure of feed efficiency [43,97]. They described that the predominant families, including Turicibacteraceae, Ruminococcaceae, and Enterobacteriaceae, were more affected by restriction feeding in both gut sections regardless RFI. Moreover, network analysis revealed that few taxa were associated with productivity level compared with the impact of nutrient availability on the microbiota richness.

The evidence described above involves birds sacrifice, thus the monitoring of changes in the microbiota over time implies the use of other birds, which introduces a greater variability in the determinations. Despite the advantages of cloacal swabbing as sampling method, the microbiota obtained from cloacal samples does not always reflect the diversity observed in intestines segments [23,101]. Stanley et al. showed that the use of fecal swab as sampling method may be useful for qualitative monitoring of the cecal microbiota, but they mention that it is necessary to study a large number of samples since the emptying of different regions of the gastrointestinal tract introduce significant variability in the data [102]. Other authors compared the composition of fecal samples in groups with different efficiency and detected similar abundances of *Lactobacillus* and *Bacteroides* regarding performance level [92]. *Acinetobacter*, *Anaerosporobacter*, and *Arcobacter* were abundant in the lower efficiency group whereas Enterobacteriaceae and Faecalibacterium were prevalent in the better efficiency group. However, there is evidence that the relative abundance of some bacteria in feces does not correlate with the composition obtained from small or large intestines [10,36]. Therefore, those results should be taken with caution to compare or evaluate performance. 

Gut microbiota and avian productivity are closely linked, and its association has been widely studied. However, the findings are sometimes contradictory or non-conclusive and it is difficult to identify specific bacterial populations that could reproducibly improve productivity and modulate the microbiota to a desired one, considering that the cause/effect relationship is still unclear. Further studies are needed to develop innovative tools and technologies that could contribute to enhance non-invasively gut microbiota monitoring.

## 5. Conclusions and Perspectives

Gut microbiota is a complex and dynamic system with multifaceted relationships between gut microbial communities and host immune system components. It clearly must be studied as complex systems to produce robust conclusions. However, it is becoming more obvious that both microbiota and immunity are key drivers for productivity. The very first regulation of the immune system associated with constant sensing of microbes implies that indirect changes in this conditioning may have long-term consequences on the capacity of the host to mount defenses and develop inflammatory or immune conditions that would have impact in animal productivity. Alteration of the composition and function of the microbiota by diets, environmental conditions, or use of antibiotics or other strong antimicrobial compounds can transform our microbial allies into potential liabilities.

The increasing use of technology and automation in chicken production establishments provides the possibility of monitoring in real time many variables in the poultry houses including physicochemical factors such as temperature, water pH, bed humidity and ammonium concentration in the air, and also bird factors such as determination of physiological parameters, imaging technologies to follow flock movements and infrared sensors to evaluate birds′ thermoregulatory features [103], all of which can help to quickly detect welfare, health, and management problems in the farms. Given the increasing evidence demonstrating the importance of the microbiota in maintaining intestinal homeostasis, and tools available to modulate it with impact in chicken productivity, it is likely that monitoring the composition of gut microbiota in the farms could be routinely included in the future.

## Figures and Tables

**Figure 1 microorganisms-07-00374-f001:**
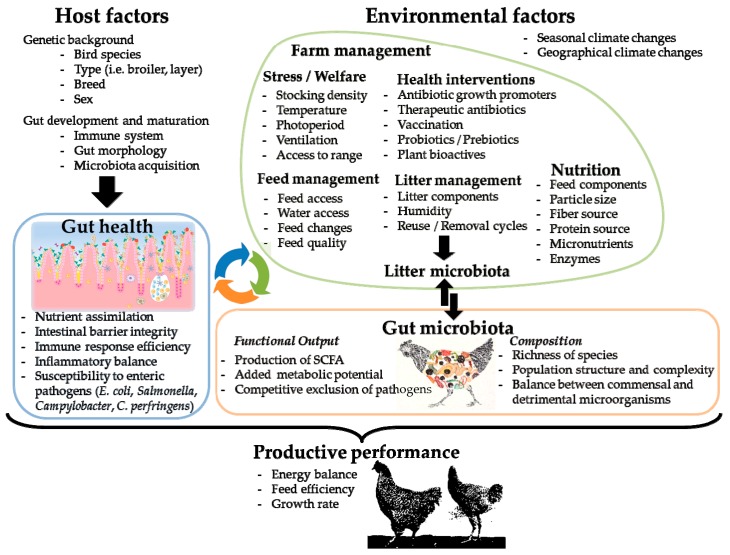
Key environmental and host-related factors that shape chicken gut microbiota and its interplay with gut health and productive performance.

**Table 1 microorganisms-07-00374-t001:** Microbial taxa associated with high and low productivity in chickens.

Sample	Performance Parameter	Microbial Taxa Identified		Ref.
		High Productivity	Low Productivity	
Crop	BW	Bacteroidetes, Euryarchaeota, *Ruminococcus*, *Faecalibacterium*	Actinobacteria, *Bifidobacterium*, *Lactobacillus*	[93]
	BW	*Clostridium coccoides*	Enterobacteria, *E. coli*, *Shigella*	[94]
Duodenum	RFI	*Lactobacillus*	*Bacteroides*	[10]
Jejunum	FCR	No differences	[91]
Ileum	FCR	*E. coli*, *Gallibacterium anatis*	*L. salivarius*, *L. aviarius*, *L. crispatus*	[90]
	BW	Euryarchaeota, *Spirochaetes*, *Bifidobacterium*, *Methanobrevibacter*	*Streptococcus*, *Akkermansia*	[93]
	BW	*Bacteroides*		[94]
	RFI, TBWG, TFI	Enterobacteriaceae	*Lactobacillus*, *Ruminococcus*, *Turicibacter*	[43]
Ileum—Cecum	RFI	*Turicibacter*, *Ruminococcus*, *Coprococcus*	Clostridiales, Proteobacteria	[36]
	BW	Firmicutes, *Lactobacillus*, Tenericutes, Actinobacteria	Firmicutes, Proteobacteria, Bacteroidetes, *Clostridium*	[89]
	BW	*Bacteroides*, *Bilophila*, *Butyricimonas*, *Faecalibacterium*	*Anaerotruncus*, *Bacteroides*, *Clostridium*, *Coprobacillus*, *Coprococcus*, *Enterococcus*, *Lactobacillus*, *Staphylococcus*, *Ruminococcus*, *Streptococcus*, unclassified Enterobacteriaceae	[23]
Cecum	FCR	*B. fragilis*, unknown bacteria	*Ruminococcus*, *L. crispatus*, Clostridiales, unknown bacteria	[91]
	RFI	*Akkermansia*, *Prevotella*, *B. coprophilus*, *L. delbrueckii*, *Veillonella dispar*, *L. reuteri*, *Prochlorococcus marinus*	*F. prausnitzii*, *Parabacteroides distasonis*, *Thermobispora bispora*, *Helicobacter*	[10]
	FCR	*F. prausnitzii*, *C. lactatifermentans*, *R. torques*	*B. vulgatus*, *Alistipes finegoldii*	[90]
	BW	*Lactococcus*	Lentisphaerae, Verrucomicrobia, *Akkermansia*, *Anaerovibrio*, *Prevotella*	[93]
	BW	*Lactobacillus*	*Escherichia* / *Shigella*	[94]
	BW		*Campylobacter*	[95]
	AME, FCR, GE, GR	Lachnospiraceae, Ruminococcaceae, Erysipelotrichaceae, Catabacteriaceae, *Ruminococcus*, *Faecalibacterium*, *Clostridium*	*Lactobacillus*, *Clostridium*	[96]
	RFI, TBWG, TFI		*Anaerotruncus*, Enterobacteriaceae, *Ruminococcus*, Clostridiales	[43]
Feces	RFI	Lachnospiraceae, *Dorea*	*Lactobacillus*, *Acinetobacter*	[36]
	BW	No differences		[89]
	RFI	*Helicobacter*	*Lactobacillus*, *Clostridium*	[10]
	RFI, TBWG, TFI	Enterobacteriaceae, Lactobacillaceae	Comamonadaceae, Moraxellaceae, *Acinetobacter*	[97]
	FCR	Enterobacteriaceae, Victivallaceae, Synergistaceae, Prevotellaceae, Rikenellaceae, Ruminococcaceae	Fusobacteriaceae, Flavobacteriaceae, Rhizobiaceae, Vibrionaceae, Xanthomonadaceae, Comamonadaceae, Campylobacteraceae, Incertae Sedis XIII	[92]
	FCR	Lactobacillaceae, Bacteroidaceae: no differences		[92]
	RFI, TBWG, TFI	*L. salivarius*, *L. crispatus*, *Anaerobacterium*	*Klebsiella*	[98]

AME: apparent metabolizable energy; BW: body weight; FCR: feed conversion ratio; GE: gross energy; GR: gain rate; RFI: residual feed intake; TBWG: total body weight gain; TFI: total feed intake.

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
