# Peer review of "Microbiota, Gut Health and Chicken Productivity: What Is the Connection?"

_microorganisms, 2019, doi:10.3390/microorganisms7100374_

Round 1
Reviewer 1 Report
This is a good review of the state of microbiota knowledge in the poultry industry. The writing is clear and concise, but this reviewer believes that some reorganization can help improve the flow and impact of this work. Additionally, there is a disparity in the thoroughness of the review among sections. Some sections are very well researched, whereas others recieve a cursory treatment. Finally, the review misses opportunities to draw attention to remaining technical, technological, and biological challenges germane to the field of poultry gut health.
Here are a few suggested revisions:
This reviewer thinks a short section summarizing the correlation of microbiota in different regions of the GI tract, and if and how they impact health and productivity will be valuable to the reader. I note that Table 1 achieves this for production, so it would be quite valuable to have a similar summary for health and immunity. I appreciate the attention to detail on the literature of the immune health, but it is frequently debated whether there are objective criteria for health (especially in poultry), and a homeostatic stage/process is not a better indicator. Following from that, the authors should expand on immune and metabolic homeostasis in relation to the microbiota. Admittedly there is limited investigations on this front, but this review is an opportunity to draw attention to this. Hieke et al (cited, #45) showed that photoperiods are not only a management practice affecting microbiota, but also that they modulate the acquisition of the microbiota. In light of this (no pun intended), the reviewer suggests adding this new knowledge to the section 1.1 The statement on line 333, regarding the fecal microbiota being qualitatively representative of cecal microbiota is loaded with problems. This reviewer suggests qualifying this statement, or in making this more accurate in other ways. In this same section, I believe there is an opportunity to call for innovation and need for new approaches and technologies to non-invasively monitor gut microbiota in poultry. "technologization" is not a real word. Please replaced in conclusions: "balanced gut microbiota" is a vage statement and not really defined. Therefore, it is suggested that the review utilize better, quantifiable terms to represent the balance, such as homeostasis.Author Response
Response to Reviewer 1 Comments
Comments and Suggestions for Authors
This is a good review of the state of microbiota knowledge in the poultry industry. The writing is clear and concise, but this reviewer believes that some reorganization can help improve the flow and impact of this work. Additionally, there is a disparity in the thoroughness of the review among sections. Some sections are very well researched, whereas others receive a cursory treatment. Finally, the review misses opportunities to draw attention to remaining technical, technological, and biological challenges germane to the field of poultry gut health.
Here are a few suggested revisions:
A1) This reviewer thinks a short section summarizing the correlation of microbiota in different regions of the GI tract, and if and how they impact health and productivity will be valuable to the reader.
Response: In line with the comments of the reviewer, Table 1 was reordered according to GI tract sections to facilitate its interpretation. The accompanying text was restructured in the same way.
A2) I note that Table 1 achieves this for production, so it would be quite valuable to have a similar summary for health and immunity.
Response: We believe that there is not enough new bibliography about the link between chicken gut microbiota, gut immunity/health and chicken productivity. This topic has been addressed in recent reviews, which we cite in the manuscript, and part of this information has been also covered in a review within the same special issue "Gut Health in Poultry Production" (Broom, 2019 - DOI 10.3390/microorganisms7050139).
A3) I appreciate the attention to detail on the literature of the immune health, but it is frequently debated whether there are objective criteria for health (especially in poultry), and a homeostatic stage/process is not a better indicator. Following from that, the authors should expand on immune and metabolic homeostasis in relation to the microbiota. Admittedly there is limited investigation on this front, but this review is an opportunity to draw attention to this.
Response: We agree with this vision. In fact the definition of “gut health” is questioned at the beginning of the chapter. We have improved this chapter and we have divided it into two new sections (2.1 and 2.2), the first focused on the host-microbe immune interactions during development and the second on the role of gut microbiota in immune homeostasis.
A4) Hieke et al (cited, #45) showed that photoperiods are not only a management practice affecting microbiota, but also that they modulate the acquisition of the microbiota. In light of this (no pun intended), the reviewer suggests adding this new knowledge to the section 1.1.
Response: What the reviewer points out is true, the photoperiod can modulate the acquisition of the microbiota and is also a management practice. But since this review is mainly focused on highlighting factors that can alter the microbiome in commercial chicken farms, we think it is more appropriate to maintain this reference in section 1.3.
A5) The statement on line 333, regarding the fecal microbiota being qualitatively representative of cecal microbiota is loaded with problems. This reviewer suggests qualifying this statement, or in making this more accurate in other ways.
Response: The authors of the cited article discuss this issue stating that: “The emptying of different GIT areas may influence fecal profile but fecal analysis still remains a useful and powerful approach for microbiota studies in animals with proven success record in human and mammalian studies”. “Perhaps a major lesson to be learnt is that it may be misleading to draw conclusions from just a few samples; a large number of samples will represent a range of different emptying events from different regions of the GIT and may allow a more complete and representative overall picture to emerge of microbiota composition”. In order to clarify this concept, we have rewritten the sentence highlighting this limitation when using fecal swab samples.
A6) In this same section, I believe there is an opportunity to call for innovation and need for new approaches and technologies to non-invasively monitor gut microbiota in poultry.
Response: We added a sentence according to reviewer’s suggestion
A7) "Technologization" is not a real word.
Response: Corrected.
A8) Please replace in conclusions: "balanced gut microbiota" is a vage statement and not really defined. Therefore, it is suggested that the review utilize better, quantifiable terms to represent the balance, such as homeostasis.
Response: The sentence was modified according to reviewer's comment.
Reviewer 2 Report
This review deals with the chicken productivity and microbiota composition. When I started with reading, the review appeared as clearly presented. However, when reading further and further, I was lost, I did not want to continue with reading. The reason is that authors correctly presented published information but did not entered the review with their own opinion, what is the most likely correct and what are likely experimental errors or consequences of very specific experimental design. I also enjoy when the reviews are challenging and provocative, when the reviews force me think, to agree or disagree. This review only presented current data and before I started with reading, I knew that microbiota might be important for gut health and when I finished, I remained at the same knowledge – yes, gut microbiota influence gut health and chicken performance, and that more research is needed.
Well, I do not say that this review cannot be accepted for publication, but I simple say that my style and preferences are different. But it might be of interest to those entering the field and many researchers may refer to this manuscript when writing introductions to their own manuscripts.
Specific points
line 2,3, why quotations in the title?
33, delete “diseases” 48, replace “microbiome” with “microbiota”. Microbiome means all genes encoded by all bacteria forming total population, i.e. sum of individual genomes. Microbiota means total bacterial cells forming the population. Since you talk about microbes in your review, in all place replace microbiome with microbiota. 71, check recent paper to realise more on the development of chicken gut microbiota - Kubasova et al. Contact with adult hen affects development of caecal microbiota in newly hatched chicks. PLoS One. 2019;14:e0212446.l.89, 3 weeks of life. Well, though you correctly refer to studies saying this, please delete and avoid. Chickens reach sexual maturity around week 18 of life and those in wild may live for 15 -20 years. You may also check the following paper from the same group as above suggesting much longer time required for microbiota development Videnska et al. Succession and replacement of bacterial populations in the caecum of egg laying hens over their whole life. PLoS One 2014;9:e115142. And confront this with the fact that if there is a source, the hen, the development can be as fast as one day. This is what I mentioned in my intro paragraph. You should warn here that although many people report on 3 weeks to complete gut microbiota development, this is caused mainly by the fact that they worked with broilers and did not proceed with monitoring of gut microbiota beyond day 35.
l.147, I wanted to look at the reference 32 and it is your poster.
l.161, there are no subtitles in chapter 2 and this makes the text quite complicated to follow,
l.201,202,203, loose using of terms microbiome and microbiota, correct to microbiota
l.217-219, the sentence starts with bacitracin but ends up with “antibiotcs”. Which antibiotics?
l.222, beta-defensins and TLR4 are not cytokines
239, delete “(reviewed by Bloom and Kogut, 2018)”l.242-247, extremely long sentence difficult to follow
l.255, Proteobacteria, well, although these are always present in chicken gut microbiota, delete anything on being “classic”. In comparison with Bacteroidetes, Proteobacteria are microbiota members of negligible importance. In fact, the whole phylum Bacteroidetes you mentioned in your review only marginally though Bacteroidetes and Firmicutes are the two main phyla forming chicken gut microbiota.
258, if you say that Proteobacteria correlated with IL6 expression, well, and which of Proteobacteria? E. coli, Shigella, Citrobacter, Salmonella etc? And when you then specify at line 261 that you meant Salmonella, well, this is a pathogen, you cannot mix up between commensals and pathogens without any warning.l.266, without any warning you start a new sentence with In particular, the genus Faecalibacterium... This is not possible since the whole previous paragraph was on Proteobacteria. Faecalibacterium is something different.
l.271,272, back to my general criticism, so above you said that Firmicutes act anti-inflammatory and you announce, without any reference, that some Firmicutes positively correlate with inflammation. So what I should learn from this?
l.279-284, this is true but has nothing to do with microbiota. Delete.
295, Table 1. This table is useless since for many taxons it provides clearly conflicting results and authors do not help the reader at. Reader, help yourselves. You have to provide some guide, otherwise I do not know how to understand the whole business.l.299, changes were conducted, please reword
l.302,303, OK but was it associated with better or worse chicken performance, this is with what this sentence starts.
l.313, cecal
314, delete “and its association with performance parameters” 315, if you find something in 1 out of 3 trials, it is better not to talk about since in two cases this finding was not truel.322, ileal
l.333,334, authors do not understand physiology of chicken intestinal tract. Faecal material can be either representing small intestine microbiota if you collect faecal material which passed directly from ileum to colon. Or you can collect caecal excretions, what happens usually twice a day when caecal contents are voided from the caecum to colon. And of course, just the following faecal material passing from the ileum to colon will be contaminated by remains of caecal excretion and therefore this material will be something in between ileal and caecal microbiota. What you say is not correct.
l.342, I do not understand why you decided for this, a paragraph of 6 lines, I recommend to delete it completely.
l.346, what positive trend was observed??? I have no idea what you wanted to say.
l.348, except for general statements, there are no specific conclusions in whole this chapter
Author Response
Response to Reviewer 2 Comments
Comments and Suggestions for Authors
This review deals with the chicken productivity and microbiota composition. When I started with reading, the review appeared as clearly presented. However, when reading further and further, I was lost, I did not want to continue with reading. The reason is that authors correctly presented published information but did not entered the review with their own opinion, what is the most likely correct and what are likely experimental errors or consequences of very specific experimental design. I also enjoy when the reviews are challenging and provocative, when the reviews force me think, to agree or disagree. This review only presented current data and before I started with reading, I knew that microbiota might be important for gut health and when I finished, I remained at the same knowledge – yes, gut microbiota influence gut health and chicken performance, and that more research is needed.
Well, I do not say that this review cannot be accepted for publication, but I simple say that my style and preferences are different. But it might be of interest to those entering the field and many researchers may refer to this manuscript when writing introductions to their own manuscripts.
Specific points
B1) L2,3 -Why quotations in the title?
Response: Quotation marks were removed.
B2) L33 - Delete “diseases”
Response: Deleted.
B3) L48 -Replace “microbiome” with “microbiota”. Microbiome means all genes encoded by all bacteria forming total population, i.e. sum of individual genomes. Microbiota means total bacterial cells forming the population. Since you talk about microbes in your review, in all place replace microbiome with microbiota.
Response: We understand that the use of the term microbiome can bring some confusion as there are different definitions of it. A commonly used definition is that mentioned by the reviewer, which considers the microbiome as the sum of all microbial genomes present in an environment. However, that is also the definition of metagenome, and many authors refer to the microbiome as “the entire habitat, including the microorganisms (bacteria, archaea, lower and higher eurkaryotes, and viruses), their genomes (i.e., genes), and the surrounding environmental conditions” unlike the term microbiota, which refers to “the assemblage of microorganisms present in a defined environment” (Marchesi, J.R.; Ravel, J. “The vocabulary of microbiome research: a proposal”. Microbiome. 2015. DOI: 10.1186/s40168-015-0094-5). For this reason, in certain parts of the review we used the term microbiome to refer not only to the set of microorganisms that make up the gut microbiota but also the surrounding environment, including the intestinal epithelium and the host's immune system components.In order to avoid any confusion, we have eliminated the word microbiome throughout the manuscript using microbiota instead, as requested by the reviewer.
B4) L71 -Check recent paper to realise more on the development of chicken gut microbiota - Kubasova et al. Contact with adult hen affects development of caecal microbiota in newly hatched chicks. PLoS One. 2019;14:e0212446.
Response: This is an interesting paper. In fact, the results are consistent with the concept we introduced in this section regarding certain groups of Firmicutes that can be acquired from environmental sources instead of from the hens. We have added the citation of this article at the end of the paragraph.
B5) L89 - 3 weeks of life. Well, though you correctly refer to studies saying this, please delete and avoid. Chickens reach sexual maturity around week 18 of life and those in wild may live for 15 -20 years. You may also check the following paper from the same group as above suggesting much longer time required for microbiota development Videnska et al. Succession and replacement of bacterial populations in the caecum of egg laying hens over their whole life. PLoS One 2014;9:e115142. And confront this with the fact that if there is a source, the hen, the development can be as fast as one day. This is what I mentioned in my intro paragraph. You should warn here that although many people report on 3 weeks to complete gut microbiota development, this is caused mainly by the fact that they worked with broilers and did not proceed with monitoring of gut microbiota beyond day 35.
Response: We have modified the sentence to highlight that reference is made to broilers in commercial conditions, since the review is mainly focused in this production. Citations were modified accordingly. We have also improved the subsequent sentence by clarifying that development of the microbiota can vary depending on the genetic background of the birds and management factors previously reviewed elsewhere. We also included as an example a citation to the paper suggested by the reviewer about the succession of species in laying hens.
B6) L147 - I wanted to look at the reference 32 and it is your poster.
Response: Citation number 32 does not refer to a poster but to a conference recently given at the symposium "Gut Health in Production of Food Animals 2018". The abstract can be accessed through the proceedings of the event. A paper with the analysis of these results is in full preparation and will be submitted for publication in the next few weeks. If the reviewer considers it more pertinent, it could be cited as unpublished data.
B7) L161 - There are no subtitles in chapter 2 and this makes the text quite complicated to follow.
Response: We have divided this chapter into two sections (2.1 and 2.2), the first focused on the host-microbe immune interactions during development and the second on the role of the microbiota in immune homeostasis.
B8) L201-203 -Loose using of terms microbiome and microbiota, correct to microbiota
Response: We have reformulated the sentence to use only the term microbiota.
B9) L217-219 -The sentence starts with bacitracin but ends up with “antibiotcs”. Which antibiotics?
B10) L222 -Beta-defensins and TLR4 are not cytokines
Response: Deleted.
B11) L239 -Delete “(reviewed by Bloom and Kogut, 2018)”
Response: Deleted
B12) L242-247 -Extremely long sentence difficult to follow
Response: The sentence was shortened and simplified.
B13) L255 - Proteobacteria, well, although these are always present in chicken gut microbiota, delete anything on being “classic”. In comparison with Bacteroidetes, Proteobacteria are microbiota members of negligible importance. In fact, the whole phylum Bacteroidetes you mentioned in your review only marginally though Bacteroidetes and Firmicutes are the two main phyla forming chicken gut microbiota.
Response: This entire paragraph was revised and rewritten since some sentences were mistakenly copied from the original article corresponding to citation number 28 (Oakley and Kogut 2016)and others were simply misinterpreted. This sentence was completely removed.
B14) L258 -If you say that Proteobacteria correlated with IL6 expression, well, and which of Proteobacteria? E. coli, Shigella, Citrobacter, Salmonella etc? And when you then specify at line 261 that you meant Salmonella, well, this is a pathogen, you cannot mix up between commensals and pathogens without any warning.
Response: We included this sentence in order to clarify “Although no specific genera within the Proteobacteria were significantly correlated with cytokine expression, the most abundant genera within phylum Proteobacteria were classified as Escherichia/Shigella”
B15) L266 -Without any warning you start a new sentence with In particular, the genus Faecalibacterium... This is not possible since the whole previous paragraph was on Proteobacteria. Faecalibacterium is something different.
Response: This was a simple mistake. It belongs to the Firmicutes. Corrected.
B16) L271,272 - Back to my general criticism, so above you said that Firmicutes act anti-inflammatory and you announce, without any reference, that some Firmicutes positively correlate with inflammation. So what I should learn from this?
Response: We have rephrased this whole sentence to emphasize that not all Firmicutes behave in the same way regarding the inflammatory response induced.
B17) L279-284 - This is true but has nothing to do with microbiota. Delete.
Response: Deleted. Part of this paragraph was used below to highlight the sources of variability that may influence the microbiota studies summarized in Table 1.
B18) L295 - Table 1. This table is useless since for many taxons it provides clearly conflicting results and authors do not help the reader at. Reader, help yourselves. You have to provide some guide, otherwise I do not know how to understand the whole business.
Response: Table 1 was reordered according to the sampling site to facilitate its interpretation. The text was restructured in the same way. Part of the general conclusions were transferred at the end of this chapter, highlighting the great variability that exists in the literature regarding taxa associated with better or worse chicken productivity.
B19) L299 - Changes were conducted, please reword
Response: Changed to “driven”.
B20) L302,303 - OK but was it associated with better or worse chicken performance, this is with what this sentence starts.
Response: The sentence regarding ileal microbiota was completely rewritten.
B21) L313 - Cecal
Response: Corrected.
B22) L314 -Delete “and its association with performance parameters” 315, if you find something in 1 out of 3 trials, it is better not to talk about since in two cases this finding was not true
Response: The sentence was deleted according to the reviewer's recommendation.
B23) L322 - ileal
Response: Corrected.
B24) L333, 334 -Authors do not understand physiology of chicken intestinal tract. Faecal material can be either representing small intestine microbiota if you collect faecal material which passed directly from ileum to colon. Or you can collect caecal excretions, what happens usually twice a day when caecal contents are voided from the caecum to colon. And of course, just the following faecal material passing from the ileum to colon will be contaminated by remains of caecal excretion and therefore this material will be something in between ileal and caecal microbiota. What you say is not correct.
Response: The authors of the cited article discuss this issue stating that: “The emptying of different GIT areas may influence fecal profile but fecal analysis still remains a useful and powerful approach for microbiota studies in animals with proven success record in human and mammalian studies”. “Perhaps a major lesson to be learnt is that it may be misleading to draw conclusions from just a few samples; a large number of samples will represent a range of different emptying events from different regions of the GIT and may allow a more complete and representative overall picture to emerge of microbiota composition”.In order to clarify this concept, we have rewritten the sentence highlighting this limitation when using fecal swab samples.
B25) L342 - I do not understand why you decided for this, a paragraph of 6 lines, I recommend to delete it completely.
B26) L346 - What positive trend was observed??? I have no idea what you wanted to say.
Response: This paragraph was deleted according to the reviewer's recommendation.
B27) L348 -Except for general statements, there are no specific conclusions in whole this chapter.
Response: Part of the general conclusions were transferred at the end of this chapter, highlighting the great variability and inconsistent results that exists in the literature regarding taxa associated with better or worse chicken productivity.
Round 2
Reviewer 2 Report
Revised version of the manuscript is better than the original submission and though I still may have some reservations, I recommend the manuscript for publication.